# A Fine-Tuning Approach to Belief State Modeling

**Samuel Sokota**[*]
Carnegie Mellon University
ssokota@andrew.cmu.edu

**Hengyuan Hu**
Meta AI
hengyuan@fb.com

**David J. Wu**
Meta AI
dwu@fb.com

**J. Zico Kolter**
Carnegie Mellon University
zkolter@cs.cmu.edu

**Jakob Foerster**
Oxford University
jakob.foerster@eng.ox.ac.uk

**Noam Brown**
Meta AI
noambrown@fb.com

## Abstract

We investigate the challenge of modeling the belief state of a partially observable Markov system, given sample-access to its dynamics model. This problem setting is often approached using parametric sequential generative modeling methods. However, these methods do not leverage any additional computation at inference time to increase their accuracy. Moreover, applying these methods to belief state modeling in certain multi-agent settings would require passing policies into the belief model—at the time of writing, there have been no successful demonstrations of this. Toward addressing these shortcomings, we propose an inference-time improvement framework for parametric sequential generative modeling methods called belief fine-tuning (BFT). BFT leverages approximate dynamic programming in the form of fine-tuning to determine the model parameters at each time step. It can improve the accuracy of the belief model at test time because it specializes the model to the space of local observations. Furthermore, because this specialization occurs after the action or policy has already been decided, BFT does not require the belief model to process it as input. As a result of the latter point, BFT enables, for the first time, approximate public belief state search in imperfect-information games where the number of possible information states is too large to track tabularly. We exhibit these findings on large-scale variants of the benchmark game Hanabi.

## 1 Introduction

A Markov system is a sequential process in which future events are independent of past events, conditioned on the current Markov state (Gagniuc, 2017). If the Markov state cannot be directly observed, the system is said to have partial observability. This work considers three kinds of Markov systems with partial observability: hidden Markov models (HMMs), partially observable Markov decision processes (POMDPs), and factored observation stochastic games (FOSGs). An HMM is a Markov system in which, at each time step, a distribution determined by the current (unobservable) Markov state generates an observable and a new (unobservable) Markov state (Rabiner & Juang, 1986). A POMDP is a generalization of an HMM in which an agent influences the trajectory of the system by taking actions (Kaelbling et al., 1998). A FOSG is a generalization of a POMDP in which multiple agents receive observations from the system and take actions that influence the trajectory of the system (Kovařík et al., 2022).

An important inference problem associated with Markov systems with partial observability is belief state modeling. The objective of belief state modeling is to compute the posterior (called the belief state) over the current Markov state. This problem is important because belief states are sufficient information to forecast future events and to anticipate how the system will respond to external actors. In sufficiently small systems, the problem can be solved exactly using the classical forward algorithm

---

[*]Work done while at Meta AI.

(Rabiner, 1989), which is based on tabular dynamic programming. However, in larger systems, the forward algorithm is inapplicable, as it scales quadratically in the number of Markov states.

In such cases, an appealing alternative is to learn an approximate model of the belief state using a parametric sequential generative model. Parametric models are appealing because they can be scaled to very large settings and can be trained from samples. However, we suggest that naively performing inference on parametric models has two significant drawbacks. First, doing so does not leverage any additional computation at inference time to improve the accuracy of the model. While the idea of performing additional local improvement is widely leveraged in reinforcement learning under the terms decision-time planning and search (Silver et al., 2018; Schrittwieser et al., 2020; Brown et al., 2020a), it has gone largely overlooked in the context of approximating belief states in large systems. Second, in the context of public belief state modeling in multi-agent systems, parametric models require a representation of the policy as input at each time step. At the time of writing, there has been no successful demonstration of this. As a result, performing belief modeling in large multi-agent systems remains out of reach, rendering celebrated algorithms for imperfect information games inapplicable (Brown & Sandholm, 2017; Moravčík et al., 2017; Brown & Sandholm, 2019).

To address these shortcomings, we propose an inference-time improvement framework for parametric models called belief fine-tuning (BFT). At each time step, BFT uses the belief model for the current step to generate an empirical distribution of current Markov states. Next, it uses this empirical distribution, along with the dynamics model (and player policies), to generate an empirical distribution of next Markov states and observations. Finally, BFT fine-tunes the belief model for the next time step using the latter empirical distribution. BFT can improve the accuracy of a belief model, even if it has been trained to convergence, by specializing its capacity to the space of local observations (Silver et al., 2008). Furthermore, BFT can model belief states even without taking actions or policies as input because, during fine-tuning, the action or policy has already been fixed.

To demonstrate the efficacy of BFT, we proceed with our experimental agenda in two parts, focusing on the cooperative imperfect-information game Hanabi (Bard et al., 2020). First, we verify that, as advertised above, BFT can improve the accuracy of a belief model in various settings. Second, we investigate the performance of decision-time planning running on top of BFT. We find that in cases in which tracking exact belief states is tractable, BFT can yield performance competitive with that of an exact belief. Furthermore, in cases in which tracking the exact belief state is intractable, we find that performing search on top of BFT can yield substantial improvements over not performing search. This is the first instance of successful approximate public belief state-based search in a setting in which computing an exact belief state is intractable.

## 2  BACKGROUND AND NOTATION

We describe three formalisms for Markov models with partial observability, and then present a unifying notation for them.

### 2.1  HIDDEN MARKOV MODELS

An HMM is a tuple $\langle \mathcal{T}, \mathcal{O} \rangle$ where $\mathcal{T} \colon \mathbb{W} \to \Delta \mathbb{W}$ is the transition function and $\mathcal{O} \colon \mathbb{W} \to \Delta \mathbb{O}$ is the observation function, where $\Delta \mathbb{O}$ is the simplex on $\mathbb{O}$. At each time, the current Markov state $w^t$ generates a new Markov state $W^{t+1} \sim \mathcal{T}(w^t)$. The new Markov state generates a new observation $O^{t+1} \sim \mathcal{O}(W^{t+1})$. The belief state is the posterior $\mathcal{P}(W^t \mid o^1, \dots, o^t)$ over the Markov state, given the history of observations.

### 2.2  PARTIALLY OBSERVABLE MARKOV DECISION PROCESSES

A POMDP is a tuple $\langle \mathcal{T}, \mathcal{O}, \mathcal{R} \rangle$ where $\mathcal{T} \colon \mathbb{W} \times \mathbb{A} \to \Delta \mathbb{W}$ is the transition function, $\mathcal{O} \colon \mathbb{W} \to \Delta \mathbb{O}$ is the observation function, and $\mathcal{R} \colon \mathbb{W} \times \mathbb{A} \to \mathbb{R}$ is the reward function. At each time step, an agent selects an action $a^t \in \mathbb{A}$ as a function of the history of its actions and observations $(o^1, a^1, \dots, o^t)$. This action is used in conjunction with the current Markov state $w^t$ to generate a new Markov state $W^{t+1} \sim \mathcal{T}(w^t, a^t)$ and a reward $r^t = \mathcal{R}(w^t, a^t)$. The observation for the next time step $O^{t+1} \sim \mathcal{O}(W^{t+1})$ is determined as a function of the new Markov state. The belief state is the posterior $\mathcal{P}(W^t \mid o^1, a^1, \dots, o^t)$ over the Markov state, given the history of actions and observations.

|  | General | HMM | POMDP | FOSG |
|---|---|---|---|---|
| Markov Variable | $X$ | $W$ | $W$ | $(W, S_1, \ldots, S_n)$ |
| Emission Variable | $Y$ | $O$ | $O$ | $O_{pub}$ |
| Control Variable | $Z$ |  | $A$ | $(\pi_1^t, \ldots, \pi_n^t)$ |

Table 1: The relationship between notations for belief state modeling for different settings.

## 2.3 Factored Observation Stochastic Games

A FOSG is a tuple $\langle \mathcal{T}, \mathcal{O}_{pub}, \mathcal{O}_{priv(1)}, \ldots, \mathcal{O}_{priv(n)}, \mathcal{R}_1, \ldots, \mathcal{R}_n \rangle$, where $\mathbb{T} \colon \mathbb{W} \times \mathbb{A} \to \Delta\mathbb{W}$ is the transition function, $\mathcal{O}_{pub} \colon \mathbb{W} \to \Delta\mathbb{O}_{pub}$ is the public observation function, $\mathcal{O}_{priv(i)} \colon \mathbb{W} \to \Delta\mathbb{O}_{priv(i)}$ is player $i$'s private observation function, and $\mathcal{R}_i \colon \mathbb{W} \times \mathbb{A} \to \mathbb{R}$ is player $i$'s reward function. At each time step, each agent selects an action $a_i^t \sim \pi_i^t(s_i^t)$ as a function of the history of its observations and actions $s_i^t = (o_{pub}^1, o_{priv(i)}^1, a_i^1, \ldots, o_{pub}^t, o_{priv(i)}^t)$. The joint action $a^t = (a_1^t, \ldots, a_n^t)$ is used in conjunction with the current Markov state $w^t$ to generate a new Markov state $W^{t+1} \sim \mathcal{T}(w^t, a)$ and a reward $r_i^t = \mathcal{R}(w^t, a)$ for each player. The public observation $O_{pub}^{t+1} \sim \mathcal{O}_{pub}(W^{t+1})$ and each player's private observation $O_{priv(i)}^{t+1} \sim \mathcal{O}_{priv(i)}(W^{t+1})$ are determined as functions of the concurrent Markov state. In FOSGs the relevant belief state is the posterior $\mathcal{P}(W^t, S_1^t, \ldots, S_n^t \mid o_{pub}^1, \pi_1^1, \ldots, \pi_n^1, \ldots, o_{pub}^t)$ over the current Markov state and each player's action-observation history, conditioned on the sequence of public observations and the sequence of policies. This belief state is called the public belief state.

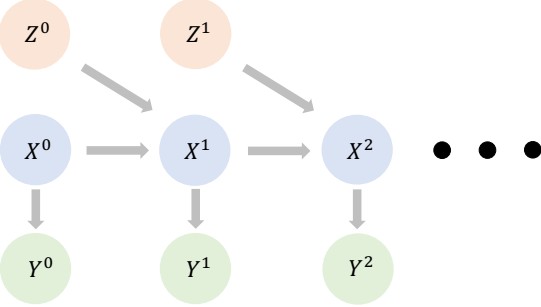

Figure 1: A graphical model depicting the belief state modeling problem. The Markov variable $X^t$ and an externally specified control variable $Z^t$ stochastically determine the next Markov variable $X^{t+1}$. Each Markov variable $X^{t+1}$ stochastically determines the contemporaneous emission variable $Y^{t+1}$. The belief state modeling problem is to approximate the posterior $\mathcal{P}(X^t \mid Y^1, Z^1, \ldots, Y^t)$ over the current Markov variable given all past and present emission variables and all past control variables.

## 2.4 General Notation for Belief State Modeling

Regardless of whether the belief state modeling problem is in the context of HMMs, POMDPs, or FOSGs, belief state modeling can be expressed as modeling a distribution $\mathcal{P}(X^t \mid Y^1, Z^1, \ldots, Y^t)$, where each $Y^j \sim \mathcal{Y}(X^j)$, each $Z^j$ is specified by an external process, and the process moves forward in time by sampling $X^{j+1} \sim \mathcal{X}(X^j, Z^j)$, as depicted in Figure 1. We call $X$ the Markov variable, $Y$ the emission variable, and $Z$ the control variable. In HMMs, the Markov variable is the Markov state $X = W$, the emission variable is the observation $Y = O$ and the control variable is null. In POMDPs, the Markov variable is the Markov state $X = W$, the emission variable is the observation $Y = O$ and the control variable is the action $Z = A$. In FOSGs, the Markov variable is a tuple of the Markov state and each player's action-observation history $X = (W, S_1, \ldots, S_n)$, the emission variable is the public observation $Y = O_{pub}$, and the control variable is the joint decision rule $Z = (\pi_1^t, \ldots, \pi_n^t)$ (i.e., the joint policy for that time step). Table 1 summarizes the relationships between these notations. To minimize clutter, we will use this more concise notation going forward. In all cases, we assume sampling access to $\mathcal{Y} \colon \mathbb{X} \to \Delta\mathbb{Y}$ and $\mathcal{X} \colon \mathbb{X} \times \mathbb{Z} \to \Delta\mathbb{X}$.

### 2.5 SEQUENTIAL GENERATIVE MODELING

The sequential generative modeling problem is to model the distribution of one sequence of variables, conditioned on another sequence of variables. Because it is a superset of the belief state modeling problem, its solution methods are immediately applicable to belief state modeling. Among the most popular of these methods are parametric modeling approaches. Parametric modeling approaches use a set of sequences $\{(X_k^0, Y_k^0, Z_k^0, \ldots, X_k^T, Y_k^T)\}_k$ collected from the dynamics model $(\mathcal{X}, \mathcal{Y})$ to minimize the difference between the output of a parameterized model $f_\theta(Y_k^0, Z_k^0, \ldots, Y_k^t)$ and the distribution over $X^t \mid Y_k^0, Z_k^0, \ldots, Y_k^t$ observed in the dataset, for each $t$ and $k$, over parameters $\theta$. They generally require two design choices. First, whether to maintain a running summary of the conditioning sequence using a recurrent network (Hochreiter & Schmidhuber, 1997) or whether to model the desired distribution as a direct function of the running sequence at each time step (Vaswani et al., 2017). Second, what kind of model to use to capture the desired distribution. Common choices include autoregressive models, adversarial models (Goodfellow et al., 2014), variational models (Kingma & Welling, 2014), or score-based models (Hyvärinen, 2005), among others. Because the experiments in this work focus on the Hanabi benchmark (Bard et al., 2020), which involves long sequences and discrete output variables, we choose to use a recurrent architecture with an autoregressive output (Sutskever et al., 2014). However, BFT is not specific to these design choices and can also be combined with other approaches for sequential generative modeling in the context of belief state modeling.

## 3 METHODOLOGY

BFT comes from the approximate dynamic programming perspective on belief state modeling. We describe this perspective below, discuss its advantages, and then introduce BFT.

### 3.1 BELIEF STATE MODELING AS APPROXIMATE DYNAMIC PROGRAMMING

A second way to approach belief state modeling—distinct from the standard sequential generative modeling setups discussed in the background—is as an approximate dynamic programming problem. Observe, for some fixed control variable $z^{t-1}$, previous belief state $b^{t-1}$, and current emission variable $y^t$, we have

$$b_{y^t}^t = \mathcal{P}(X^t \mid y^t, z^{t-1}, b^{t-1}) \tag{1}$$

$$\propto \mathcal{P}(y^t \mid X^t, z^{t-1}, b^{t-1})\mathcal{P}(X^t \mid z^{t-1}, b^{t-1}) \tag{2}$$

$$= \mathcal{P}(y^t \mid X^t)\mathbb{E}_{X^{t-1} \sim b^{t-1}}\mathcal{P}(X^t \mid z^{t-1}, X^{t-1}). \tag{3}$$

Line (1) follows by definition, line (2) follows by Bayes' rule, and line (3) follows by the Markov property and the law of total probability. As a result of this proportionality, the procedure

1. Sample $X^{t-1} \sim b^{t-1}$
2. Sample $X^t \sim \mathcal{X}(X^{t-1}, z^{t-1})$
3. Sample $Y^t \sim \mathcal{Y}(X^t)$

yields a sample $X^t \sim b_{Y^t}^t$. Thus, given sample-access to $\mathcal{X}, \mathcal{Y}$, and the distribution of $X^0$, it is entirely possible to train a belief model inductively via approximate dynamic programming, whereby the belief states at time step one bootstrap from the initial belief state, the belief states at time step two bootstrap from the belief states at time step one, and so forth, as shown in Figure 2.

While this procedure is known, it is unusual for it to be used to train belief state models because the training data it produces is biased—unless the approximation to $b^{t-1}$ is exactly correct, the training data for $b_{Y^t}^t$ will only come from an approximately correct distribution. In contrast, sampling full trajectories guarantees that the training data will come from exactly the correct distribution.

### 3.2 ADVANTAGES OF APPROXIMATE DYNAMIC PROGRAMMING

Despite that approximate dynamic programming produces biased samples, we argue here that the approximate dynamic programming perspective offers advantages that have gone largely overlooked in existing literature. These advantages derive from the fact that approximate dynamic programming

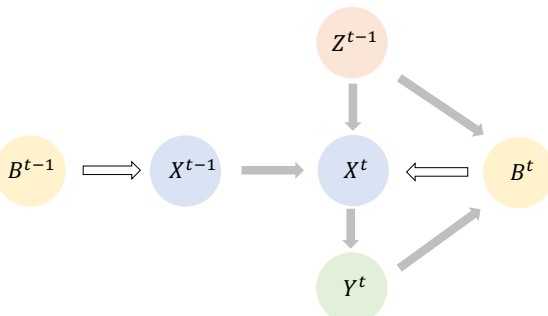

Table 2: A graphical model depiction of the dynamic programming perspective on belief state modeling. Gray arrows denote causal relationships; white arrows denote distributional modeling.

allows us to generate local data. Given previous belief state $b^{t-1}$, generating samples for the current time step is as simple as following the procedure above. In contrast, generating unbiased versions of the same samples would require sampling a collection of initial Markov states $X_1^0, \ldots, X_k^0 \sim \mathcal{X}^0$, propagating them forward using the same sequence of control variables $z^0, z^1, \ldots$, and rejecting the rollouts that yielded emission sequences $Y^0, \ldots, Y^{t-1}$ inconsistent with the previously observed sequence of emission variables. For large $t$, it becomes prohibitively unlikely to sample a trajectory with exactly the same sequence of observations.

The ability to generate local data in the context of belief state modeling may be helpful for two reasons. First, it may increase the accuracy of the belief model by allowing it to perform an additional improvement step immediately prior to inference time. Even if the model was trained to convergence offline, it may be the case that the model does not possess sufficient capacity to capture the exact belief for every possible sequence of emission and control variables. Therefore, an additional improvement step can provide utility by reallocating the belief model's capacity to the space of immediate possible futures, thereby increasing its ability to approximate the belief state for those futures. These ideas have long been understood in reinforcement learning literature (Silver et al., 2008), but are overlooked in the context of belief modeling.

A second advantage of inference-time improvement is that the control variable can be omitted from the belief model. This is a result of the fact that the Markov variable only depends on the control variable from the previous time step (not the current time step). Therefore, so long as inference-time improvement takes place after the control variable for the previous time step has been decided, the belief state depends only on the contemporaneous emission variable. In the context of HMMs or POMDPs, where the control variables are nulls and actions, respectively, the ability to omit the control variable carries little value. However, in FOSGs, where the control variables are themselves joint policies, the ability to omit the control variable is an important attribute. Indeed, at the time of writing, there have been no convincing demonstrations of belief state modeling at scale in FOSGs due to the fact that it is not clear how to train a sequential generative model as a function of policy networks.

### 3.3 BELIEF FINE-TUNING

Toward leveraging the advantages of inference-time improvement discussed above, **we propose a framework for belief fine-tuning (BFT).** BFT takes a pretrained belief model $f_\theta$ as input and, at each time step during online inference, performs the following procedure:

1. Sample $X_1^{t-1}, \ldots, X_k^{t-1} \sim f_{\theta^{t-1}}(y^0, z^0, \ldots, y^{t-1})$.
2. Sample $X_i^t \sim \mathcal{X}(X_i^{t-1}, z^{t-1})$ for each $i = 1, \ldots, k$.
3. Sample $Y_i^t \sim \mathcal{Y}(X_i^t)$ for each $i = 1, \ldots, k$.
4. Set $\theta^t \leftarrow$ fine-tune$(\theta, \{(Y_i^t, X_i^t) \mid i = 1, \ldots, k\})$

For the final step, exactly how the fine-tuning is performed depends on the structure of the architecture $f$. If $f$ takes the control variable as input, then $z^{t-1}$ should also be passed into the model during

fine tuning; if $f$ is recurrent, then the hidden state $h_\theta^{t-1}(y^0, z^0, \ldots, y^{t-1})$ should also be passed into the model; if $f$ takes the full sequence as input, then the full sequence should also be passed in during fine-tuning. In principle, BFT could produce accurate distributions from an untrained model that only took a single emission variable $Y$ as input. However, we do not recommend this approach, as it amounts to training a new generative model at each time step from scratch.

## 4 EXPERIMENTS

We divide our experimental investigation into two parts. In the first part, we explore the extent to which BFT can improve the belief quality of a parametric model in HMMs, POMDPs, and FOSGs. In the second part, we explore the idea of performing search on top of beliefs from BFT.

We use the cooperative card game Hanabi (Bard et al., 2020) for these experiments. Hanabi resembles something in the realm of a cooperative Solitaire—the players are tasked with collectively playing cards of each suit in rank order into piles on the table. However, a characterizing feature of Hanabi is that no player can observe its own cards. Instead, each must infer information about its own cards from other players' actions. A more elaborate description of Hanabi is in the appendix.

In Hanabi, if the participants in the game agree to follow a fixed joint policy, the system can be considered an HMM from the perspective of a third party who does not observe the players' cards. If all but one of the players in the game agree to follow a fixed (joint) policy, the system can be considered a POMDP from the perspective of the player with a varying policy. If none of the players' policies are fixed, then the system can be modeled as a FOSG. Therefore, Hanabi allows us to investigate all three systems within the same infrastructure.

The codebase for our experiments can be found at `https://github.com/ facebookresearch/off-belief-learning`. Additional ablations on number of fine-tuning steps, amount of offline training, network capacity, and part of network tuned are included in the appendix.

### 4.1 BELIEF FINE-TUNING FOR IMPROVING BELIEF QUALITY

For our first set of experiments, we verify that BFT can improve the accuracy of a pre-trained parametric belief model in HMMs, POMDPs, and FOSGs. For our experimental setup, we trained policies using independent R2D2 (Kapturowski et al., 2019) that collectively score around 24 out of 25 points, using the same hyperparameters as those found in (Hu & Foerster, 2020). We then trained a Seq2Seq model (Sutskever et al., 2014) close to convergence for an HMM belief state, the POMDP belief state, and the public belief state (i.e., the FOSG belief state). For the Seq2Seq model, we used the same hyperparameters as those found in (Hu et al., 2021). For the FOSG case, we did not feed the players' policy networks into the Seq2Seq model.

At evaluation time, we used RLSearch (Fickinger et al., 2021) on top of the R2D2 policies to generate the actions for the non-fixed players in each setting. In short, RLSearch performs additional fine-tuning of the policy or value network at decision time, using samples from the belief state and the same learning algorithm that was used during training time. We used the same hyperparameters as those found in the original paper, with the exception that we used the search policy at every time step, rather than sometimes using the blueprint policy (i.e., the policy computed using R2D2). This choice was made so as to highlight the amount of improvement achievable from BFT. For BFT, we fine-tuned the encoder of the belief network for 10,000 gradient steps at each decision-point using the same hyperparameters that were used for offline training.

|  | V0 Belief | Seq2Seq | Seq2Seq + BFT (Ours) |
|---|---|---|---|
| HMM | $2.08 \pm 0.01$ | $1.67 \pm 0.01$ | $1.58 \pm 0.01$ |
| POMDP | $1.73 \pm 0.01$ | $1.52 \pm 0.01$ | $1.39 \pm 0.01$ |
| FOSG | $2.12 \pm 0.01$ | $1.81 \pm 0.01$ | $1.62 \pm 0.01$ |

Table 3: Cross entropy per card with standard error for different settings, aggregated over 300 games.

Table 3 shows results for the Seq2Seq model, BFT on top of the Seq2Seq model, and a baseline called the V0 belief, which is naively from independence assumptions and the rules of the game (in

ignorance of the actual player policies). The V0 belief serves as a sanity check that the belief model is working and provides a sense of scale about cross entropy values for each setting. The scale for the POMDP setting differs because it conditions on the acting player's private information and the belief is over only one player's hand; in contrast, the belief in the HMM and FOSG settings are over both players' hands and condition on only public information.

For the HMM setting, we observe BFT yields an improvement over the belief model. This improvement comes despite the facts that 1) the belief model was trained using a stream of freshly generated data (so there is no overfitting problem), 2) the belief model was trained close to convergence, 3) there is no distribution shift at test time. This result may be evidence of the benefit of reallocating the capacity of the neural network at inference time.

For the POMDP setting, we observe that BFT yields an even larger improvement over the Seq2Seq model. While facts 1) and 2) of the above paragraph still hold true here, fact 3) does not. In particular, in the POMDP setting, there is covariate distribution shift—the acting player uses a search policy at evaluation time, rather than the blueprint policy used at training time. In principle, if the model had a sufficiently large amount of capacity and was trained to convergence, covariate distribution should not impact model performance. However, as a matter of practice, as is illustrated by the results in Table 3, covariate distribution shift can have a large effect on model performance.

Finally, in the FOSG setting, we observe that BFT induces the largest improvement. In this setting, the belief model faces not only covariate distribution shift, as it did in the POMDP setting, but also concept distribution shift, as a result of the facts that the belief model is ignorant to the control variables (the player policies) and that the distribution of control variables changes at test time because the agents are using search.

## 4.2 BELIEF FINE-TUNING FOR DECISION-TIME PLANNING

For our second set of experiments, we investigate whether the improvement in belief quality yielded by BFT leads to downstream improvements in search performance. For these experiments, we used the same hyperparameters as those found in (Fickinger et al., 2021) for the search algorithms and (Hu et al., 2021) for the Seq2Seq model. Again, we used 10,000 gradient steps on the encoder of the belief model with the same hyperparameters as offline training.

| | 5-card 8-hint Hanabi | | | | | |
| Blueprint | SPARTA Single Exact | RL Search Single Exact | SPARTA Multi Exact | RL Search Multi Exact | RL Search Multi Seq2Seq | RL Search Multi BFT (Ours) |
|---|---|---|---|---|---|---|
| 24.23 ±0.04 | 24.57 ±0.03 | 24.59 ±0.02 | 24.61 ±0.02 | 24.62 ±0.03 | 24.35 ±0.03 | 24.58 ±0.02 |

| | 5-card 2-hint Hanabi | | | | | |
| Blueprint | SPARTA Single Exact | RL Search Single Exact | SPARTA Multi Exact | RL Search Multi Exact | SPARTA Multi Seq2Seq | RL Search Multi BFT (Ours) |
|---|---|---|---|---|---|---|
| 22.99 ±0.04 | 23.60 ±0.03 | 23.61 ±0.03 | 23.67 ±0.03 | 23.76 ±0.04 | 23.35 ±0.03 | 23.69 ±0.03 |

Table 4: Expected returns with standard error, aggregated over 2000 games. RL Search Multi Exact is an upper bound on the performance we would hope for from BFT. RL Search Multi Seq2Seq is our baseline. BFT significantly outperforms the Seq2Seq model, and nearly matches the performance of exact beliefs.

We show performance results for (5-card, 8-hint) and (5-card, 2-hint) Hanabi in Table 4. We compare single and multi-agent SPARTA (Lerer et al., 2020) with exact beliefs, and single and multi-agent RLSearch (Fickinger et al., 2021) with exact beliefs, to two versions of multi-agent RLSearch with approximate beliefs. In one approximate-belief version, we used the Seq2Seq model trained offline. In the other version, we performed BFT on top of the Seq2Seq model after the search policy had been computed.

In both the 2-hint and 8-hint variants, exact-belief multi-agent search substantially improved upon the blueprint policy. Seq2Seq beliefs fell far short of this improvement, while Seq2Seq + BFT nearly matches the performance of exact beliefs. These results suggest that the improvement in cross entropy from BFT observed in the previous section has an important effect on downstream search.

We can further investigate the added value of BFT by looking at the self-assessed improvement of multi-agent RLSearch with each of the three belief types (Seq2Seq, Seq2Seq + BFT, Exact), shown in Figure 4.2. We compute the increase in expected return by performing rollouts from the public belief state to the end of the game, using both the policy determined by RLSearch and the blueprint policy. Using the exact belief (blue) corresponds to computing the true improvement, whereas any error in the approximate beliefs will spill over into the improvement computation. We observe that BFT (orange) mostly yields comparable self-assessed improvements to an exact belief, with some overestimation toward the end of the game. In contrast, the Seq2Seq belief (green) significantly overestimates the amount of improvement over the blueprint throughout the course of the game. These results help explain those found in Table 4—the Seq2Seq model is not able to properly assess policy values without BFT, leading it to believe that its search policy is making large improvements when, in fact, its improvements may be small or non-existent.

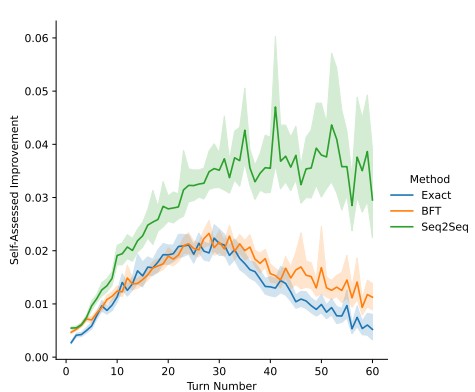

Figure 2: Without BFT, multi-agent RLSearch with Seq2Seq overestimates its performance.

| 7-card 4-hint Hanabi | | | | |
|---|---|---|---|---|
| Blueprint | RLSearch Single Seq2Seq | RL Search Single BFT (Ours) | RL Search Multi Seq2Seq | RL Search Multi BFT (Ours) |
| 23.67 ±0.02 | 24.14 ±0.04 | 24.18 ±0.03 | 23.71 ±0.07 | 24.18 ±0.03 |

Table 5: Expected returns with standard error, aggregated over 1000 games. Multi-agent search with BFT significantly outperforms the multi-agent search with a Seq2Seq belief. Computing exact beliefs is intractable in this setting.

For our final experiment, we examine 7-card Hanabi—a setting in which there is so much private information that it is difficult to compute an exact belief. For this experiment, we include single and multi-agent variants of RLSearch using approximate beliefs from Seq2Seq and BFT. Similarly to the results in Table 4, we observe in Table 5 that multi-agent RLSearch without BFT outperformed the blueprint (though not with significance in this case) but was significantly outperformed by every other setup. In contrast, multi-agent RLSearch with BFT substantially outperforms the blueprint. For single-agent RLSearch, BFT also resulted in better performance than Seq2Seq. We computed a one-sided $t$-test (computed by pairing deck orderings) and found that this improvement occurred with a $p$-value of $0.052$.

## 5 RELATED WORK

There has been much recent focus among the machine learning community on large, relatively general purpose, machine models that are fine-tuned for more specific tasks, such as BERT (Devlin et al., 2019), GPT-3 (Brown et al., 2020b), or DALL-E (Ramesh et al., 2021). Our work relates to this literature in the way it handles belief states for FOSGs—first by training a belief model ignorant of the control variable offline, then later specializing the belief model to a specific control variable during online inference, though on a much smaller scale.

More generally, the idea of improving a model from a small amount of data at inference time also places our work close to the meta-learning literature (Hospedales et al., 2021). There are various ways to perform meta-learning, such as recurrent neural networks (Ravi & Larochelle, 2017), hypernetworks (Qiao et al., 2018), or fine-tuning a network initialized using second order gradient descent (Finn et al., 2017).

BFT is also closely related to the idea of performing additional improvement at decision-time, which has a rich history in reinforcement learning literature—including many of the most widely acclaimed game-playing AI systems (Tesauro et al., 1995; Silver et al., 2018; Moravčík et al., 2017; Brown & Sandholm, 2017; 2019). However, the literature that specifically uses fine-tuning for decision-time planning is much smaller. The first works to do so with large non-linear networks (Anthony et al., 2019; Anthony, 2021) showed that a policy gradient fine-tuning-based search can perform competitively with, or even outperform, MCTS in Hex. More recently, Fickinger et al. (2021) investigated the idea of fine-tuning-based search using independent Q-learning in Hanabi and policy gradients in Ms. Pacman (Bellemare et al., 2012).

Our work also relates to literature that attempts to work with public belief states at scale in FOSGs. The Bayesian action decoder scales approximate public belief states to Hanabi using Hanabi-specific independence heuristics and an expectation propagation-like procedure (Foerster et al., 2019). While we also examine the performance of approximate belief states on Hanabi, our method differs in that it uses no independence heuristic, is focused on inference-time improvement, and is not tied specifically to Hanabi. Another paper investigating public belief states in large settings is that of Šustr et al. (2021), who propose a scalable method for approximating the value of a public belief state from samples. More precisely, they investigate a value function architecture that takes samples from public belief states as input, rather than a closed form public belief state. Our contribution is complementary to theirs, as our method aims to produce high quality public belief state samples.

Finally, our work also relates to particle filtering literature (Doucet & Johansen, 2009). Particle filtering maintains works by maintaining a set of Markov states $\{x_i\}_i$ (called particles) and corresponding weights $\{w_i\}_i$ that collectively approximate the belief state. At each time step, similarly to BFT, the collection of particles is propagated forward in time using the dynamics model. The weights are updated based on the proportional posterior probability of each particle. One disadvantage of particle filtering compared to BFT is that all of the particles may become infeasible; in other words, every particle may have probability zero given the sequence of emission and control variables. Moreover, whereas particle filtering can only resample particles from its finitely supported empirical distribution, BFT can produce arbitrary resamples.

# 6 CONCLUSION AND FUTURE WORK

In this work we introduced BFT, a framework for improving belief state models during online inference. We sought to demonstrate two claims: i) BFT can improve the performance of a belief model ii) this gain in belief accuracy yields downstream gains in search performance. We provided evidence for both claims using large scale variants of the Hanabi benchmark.

We see three lines of work worth pursuing as follow ups to this one. The first is to investigate whether incorporating meta-learning into BFT can improve its performance. While the inference-time improvement framework can be cast as a meta-learning problem, we did not actually incorporate this into the BFT objective. The second is to investigate the design of belief models that can quickly model public belief states. While BFT allows us to effectively model public belief states, it requires performing fine-tuning each time a public belief state is required. This is fine for algorithms which only use public belief states to construct the root of their search trees (Brown et al., 2018; Lerer et al., 2020), but may be too slow to use for algorithms that use public belief state-based value functions (Moravčík et al., 2017; Brown et al., 2020a; Sokota et al., 2021). The third line is to investigate eliminating the need for sampling access to the dynamics model. While MuZero (Schrittwieser et al., 2020) accomplished this for deterministic fully observable settings, determining the best way to do this for partially observable settings remains an open problem.

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

## A  APPENDIX

### A.1  RULES OF HANABI

The following rule description is quoted from the Hanabi challenge paper (Bard et al., 2020):

> Hanabi is a game for two to five players, best described as a type of cooperative solitaire. Each player holds a hand of four cards (or five, when playing with two or three players). Each card depicts a rank (1 to 5) and a colour (red, green, blue, yellow, and white); the deck (set of all cards) is composed of a total of 50 cards, 10 of each colour: three 1s, two 2s, 3s, and 4s, and finally a single 5. The goal

of the game is to play cards so as to form five consecutively ordered stacks, one for each colour, beginning with a card of rank 1 and ending with a card of rank 5. What makes Hanabi special is that, unlike most card games, players can only see their partners' hands, and not their own. Players take turns doing one of three actions: giving a hint, playing a card from their hand, or discarding a card. We call the player whose turn it is the active player.

**Hints**  On their turn, the active player can give a hint to any other player. A hint consists of choosing a rank or colour, and indicating to another player all of their cards that match the given rank or colour. Only ranks and colors that are present in the player's hand can be hinted for. To make the game interesting, hints are in limited supply. The game begins with the group owning eight information tokens, one of which is consumed every time a hint is given. If no information tokens remain, hints cannot be given and the player must instead play or discard.

**Discard**  Whenever fewer than eight information tokens remain, the active player can discard a card from their hand. The discarded card is placed face up (along with any unsuccessfully played cards), visible to all players. Discarding has two effects: the player draws a new card from the deck and an information token is recovered.

**Play**  Finally, the active player may pick a card from their hand and attempt to play it. Playing a card is successful if the card is the next in the sequence of its colour to be played. If the play is successful, the card is placed on top of the corresponding stack. When a stack is completed (the 5 is played) the players also receive a new information token (if they have fewer than eight). The player can play a card even if they know nothing about it; but if the play is unsuccessful, the card is discarded (without yielding an information token) and the group loses one life, possibly ending the game. In either circumstances, a new card is drawn from the deck.

**Game Over**  The game ends in one of three ways: either because the group has successfully played cards to complete all five stacks, when three lives have been lost, or after a player draws the last card from the deck and every player has taken one final turn. If the game ends before three lives are lost, the group scores one point for each card in each stack, for a maximum of 25.

## A.2   ADDITIONAL EXPERIMENTS

We performed additional experiments to see the effect of the number of fine-tuning steps, the number of training epochs, and the size of the belief model. Note that we did not use the same networks for these results as for the HMM results in the main body so they are not directly comparable.

### A.2.1   NUMBER OF FINE-TUNING STEPS

In the main body, we used 10,000 gradient steps for all of our experiments. In Table 6, we examine the effect of using fewer gradient steps in an HMM setting. We find that BFT can yield benefits even with as few as 100 gradient steps. We also find that the benefits appear to saturate after 1000 gradient steps. This, in part, may be because the belief model is already relatively good in the HMM case.

| V0 Belief | $2.08 \pm 0.01$ |
|---|---|
| Seq2Seq | $1.68 \pm 0.01$ |
| Seq2Seq + BFT(100 steps) | $1.64 \pm 0.01$ |
| Seq2Seq + BFT(1000 steps) | $1.61 \pm 0.01$ |
| Seq2Seq + BFT(10000 steps) | $1.61 \pm 0.01$ |

Table 6: Cross entropy per card with standard error for different settings, aggregated over 300 games.

### A.2.2 Amount of Offline Training

In the main body, our belief model was trained for 400 epochs offline. In Table 7, we examine the performance of combining BFT with lower quality belief models in an HMM setting. We find that BFT can produce large improvements even when its original model performs poorly.

| | |
|---|---|
| V0 Belief | $2.08 \pm 0.01$ |
| Seq2Seq(400 epochs) | $1.68 \pm 0.01$ |
| Seq2Seq(400 epochs) + BFT | $1.61 \pm 0.01$ |
| Seq2Seq(200 epochs) | $1.74 \pm 0.01$ |
| Seq2Seq(200 epochs) + BFT | $1.63 \pm 0.01$ |
| Seq2Seq(100 epochs) | $1.82 \pm 0.01$ |
| Seq2Seq(100 epochs) + BFT | $1.70 \pm 0.01$ |

Table 7: Cross entropy per card with standard error for different settings, aggregated over 300 games.

### A.2.3 Network Size and Part of Network Tuned

In the main body of the paper, the belief model's encoder was a two layer LSTM with 512 hidden units. Also, in the main body of the paper, we performed fine-tuning only on the encoder. In Table 8, we examine the effect of using smaller encoders and of fine-tuning the whole belief network, rather than just the encoder, in the HMM setting. We find that the benefits to performing fine-tuning may increase as the capacity of the belief network decreases, if fine-tuning is performed for the whole network. However, in the case that fine-tuning is performed only on the encoder, we did not observe this effect.

| | |
|---|---|
| V0 Belief | $2.08 \pm 0.01$ |
| Seq2Seq(512 units) | $1.68 \pm 0.01$ |
| Seq2Seq(512 units) + BFT(Encoder) | $1.61 \pm 0.01$ |
| Seq2Seq(512 units) + BFT(Full) | $1.58 \pm 0.01$ |
| Seq2Seq(256 units) | $1.76 \pm 0.01$ |
| Seq2Seq(256 units) + BFT(Encoder) | $1.65 \pm 0.01$ |
| Seq2Seq(256 units) + BFT(Full) | $1.63 \pm 0.01$ |
| Seq2Seq(128 units) | $1.87 \pm 0.01$ |
| Seq2Seq(128 units) + BFT(Encoder) | $1.84 \pm 0.01$ |
| Seq2Seq(128 units) + BFT(Full) | $1.69 \pm 0.01$ |

Table 8: Cross entropy per card with standard error for different settings, aggregated over 300 games.

### A.3 Description of Algorithm Implementation

We describe the instantiation of BFT we used for our Hanabi experiments. We used a Seq2Seq model with LSTMs as encoders and an LSTM as a decoder. In particular, our encoder took the previous control variable $z^{t-1}$, the current emission variable $y^t$, and a summary of the history $h(y^0, z^0, \ldots, y^{t-1})$ and output a next summary of the history $h(y^0, z^0, \ldots, y^t) = e(z^{t-1}, y^t, h(y^0, z^0, \ldots, y^{t-1}))$. Our decoder $d$ took the summary of the history $h(y^0, z^0, \ldots, y^{t-1}))$ as input, and autoregressively modeled the Markov state $X^t$.

1. Sample $X_1^{t-1}, \ldots, X_k^{t-1} \sim d_\theta \circ e_{\theta^{t-1}}(z^{t-2}, y^{t-1}, h_\theta(y^0, z^0, \ldots, y^{t-2}))$.
2. Sample $X_i^t \sim \mathcal{X}(X_i^{t-1}, z^{t-1})$ for each $i = 1, \ldots, k$.
3. Sample $Y_i^t \sim \mathcal{Y}(X_i^t)$ for each $i = 1, \ldots, k$.
4. Set $\theta^t \leftarrow \min_{\theta'} \text{NLL}(X_i^t, d_\theta \circ e_{\theta'}(z^{t-1}, Y_i^t, h_\theta(y^0, z^0, \ldots, z^{t-2}, y^{t-1}))$ for $i = 1, \ldots, k$.

During fine-tuning time, we first sample Markov states using the decoder with offline parameters $\theta$, the encoder using the fine-tuned parameters from the previous time step $\theta^{t-1}$, and the hidden state produced two time steps ago using the the offline parameters $\theta$. In steps 2) and 3), we propagate these samples forward to the current time step, as described in the main body of the paper. In the fourth step, we fine-tune the parameters of the encoder (with the parameters initialized as the offline

parameters $\theta$), using the decoder (with fixed offline parameters $\theta$), and the fixed hidden state from the previous time step computed using offline parameters $\theta$. The result of this fine tuning yields encoder parameters $\theta^t$. For the FOSG setting, we omitted $z$ as input in step 1) and step 4) (i.e., we only used it in step 2)).

An alternative instantiation, used for BFT(Full) in Table 8, fine-tunes both the encoder and the decoder. In this case $d_\theta$ is replaced by $d_{\theta^{t-1}}$ in step 1), and $d_\theta$ is replaced by $d_{\theta'}$ in step 4). Only fine-tuning the encoder has the advantage of preserving the equivalence between the encoding and the belief state, which may be useful for training belief state-dependent value functions in future work. However, fine-tuning the whole network can sometimes result in better performance, as we found in Table 8.

RLSearch only requires samples from the current belief state. To provide these samples, at time $t$, we generated samples from $d_\theta \circ e_{\theta^t}(z^{t-1}, y_i^t, h_\theta(y^0, z^0, \ldots, z^{t-2}, y^{t-1}))$. Again, in the FOSG case we omitted the control variables. We sampled a large number of Markov states prior to the start of search, and then passed this empirical distribution of samples to our RLSearch implementation. Note that, if fine-tuning the whole network, $d_\theta$ should be replaced with $d_{\theta^t}$ here.

