# OpenReview forum: "A Fine-Tuning Approach to Belief State Modeling"
_ICLR.cc/2022/Conference — ICLR 2022 Poster_

### Official Review · Reviewer_kvNa · 2021-11-02

**Correctness:** 4
**Technical Novelty And Significance:** 3
**Empirical Novelty And Significance:** 3
**Recommendation:** 8
**Confidence:** 4

**Main Review:**

-- After Author Response --

I am leaving my original review below for transparency, but my most significant concerns have been addressed. In particular:
- The RLSearch paper has been accepted for presentation at NeurIPS 2021.
- The authors have provided the sample sizes for their averages and they are sufficient to support their claims.

I still think the authors should aim to make the paper more self-contained by providing more algorithmic details about RLSearch (which is a new approach), how it interacts with BFT, and why it was selected for these experiments instead of a more well-established search method.

-- Original Review --

Belief state inference is an important step toward developing agents that can make sound decisions in partially observable environments. A great deal of effort has been spent studying exact inference or approximate inference with access to ground truth state distributions. This paper's focus on bringing this tool to more complex domains where representing distributions and performing exact inference are both a challenge is worthwhile and is bound to be of interest to the ICLR community. The results also fit into a larger story about neural network fine-tuning/meta-learning, which is certainly of interest.

Technically, the approach is only modestly novel, mainly applying an idea that has been studied elsewhere in a context where it hasn't been applied yet. That said, in my opinion, the empirical results are both novel and significant. In particular, the idea of directly learning a belief state model has not gained a lot of traction because it, frankly, doesn't work all that well! We can see in the 7-card Hanabi results that performing search on such a model yields essentially no benefit (we are lucky that it didn't cause harm!). The fact that the fine-tuning enabled the approximate belief model to yield a planning benefit is notable and a promising signpost toward future work, which might enable belief state inference to scale to larger, more interesting partially observable problems.

All that said, I do have a significant concern about this paper. The empirical results rely heavily on an existing algorithm called RLSearch. As far as I can tell, the paper that describes RLSearch has not been peer reviewed and has only been available on arXiv for about a month at time of reviewing. The paper makes a very brief attempt to explain RLSearch at a high level, and how that algorithm is altered for this paper, but this is not sufficient. From this paper, I do not know what algorithm is being performed in these experiments or why RLSearch is even a sound foundation to build upon, let alone why it is the right base algorithm to combine with BFT in order to answer these empirical questions. Can BFT be applied to other search methods? Is there some special synergy with RLSearch since they both appear to be fine-tuning methods? If the answers to these questions is no, then I would recommend using a baseline algorithm that is more established and better studied. If the answer is yes, then that needs to be made far more explicit and clear. If the answer is "we don't know" then I think that's a major missing piece of this work.

The fact is that the paper is not sufficiently self-contained and the parts that it doesn't contain haven't been peer reviewed. That makes me deeply uncomfortable and makes it difficult for me to confidently assess the technical quality and the significance of the findings.

Another concern I have is that I don't know how many independent trials are represented in the empirical results. Maybe I just missed it, but it seems like a very important detail to state. If the number is not sufficiently high, that would raise questions about the strength of the support for the conclusions.

A couple of more minor issues:
- The references section is, frankly, a mess. There is no consistency in the formatting and content of the references and some don't even list a venue, just authors, a title, and a year.
- p. 2: "At the time of writing, there has been no successful demonstration of this." It wasn't clear to me at this point what "this" refers to. Even now I'm not totally sure. It would be good to have a more clear expression of the open problem here.

**Summary Of The Paper:**

This paper concerns the idea of directly learning a model of a belief state MDP from samples from the ground truth dynamics and observation models. One could then use that model for decision-time planning (e.g. some form of tree search). In particular, the paper aims to improve upon this idea by fine-tuning the belief state model before search using samples from local state region. Experiments are performed in Hanabi, showing that fine-tuning does improve belief accuracy and enables search through belief space to improve control performance in problems where exact inference is impractical.

**Summary Of The Review:**

-- After Author Response --

The paper is well-written and considers an important and challenging problem. The empirical results seem to offer a path to progress in a direction that hasn't shown much promise. The paper could be improved by offering more details about RLSearch, a recently introduced approach upon which the experiments heavily rely.

-- Original Summary --

The paper is well-written and considers an important and challenging problem. The empirical results seem to offer a path to progress in a direction that hasn't shown much promise. However, the algorithmic results make heavy use of an algorithm that is not sufficiently described in this paper and that has not been peer-reviewed. Though I would really like to have these results in the literature, that makes me uncomfortable with recommending acceptance.

---

> ### Author Response · Authors · 2021-11-18
> **Thanks for the review!**
>
> > Can BFT be applied to other search methods? Is there some special synergy with RLSearch since they both appear to be fine-tuning methods? If the answers to these questions is no, then I would recommend using a baseline algorithm that is more established and better studied. If the answer is yes, then that needs to be made far more explicit and clear. If the answer is "we don't know" then I think that's a major missing piece of this work.
>
> Yes, BFT can be combined with any search method that only requires access to the current belief state. The reason that we focused on RLSearch here is because it is one of only two methods that have been shown to be scalable enough to perform multi-agent search in Hanabi. And the other such method---multi-agent SPARTA---has the downside of requiring enumerating the support of the public belief state, which is not scalable to settings with billions or more infostates.
>
> > Another concern I have is that I don't know how many independent trials are represented in the empirical results. Maybe I just missed it, but it seems like a very important detail to state. If the number is not sufficiently high, that would raise questions about the strength of the support for the conclusions.
>
> We agree, this is an important detail that should have been stated in the original submission. For the belief quality experiments, we used N = 300 games. For the 5-card decision-time planning experiments, we used N = 2000 games. For the 7-card decision-time planning experiments, we used N = 1000 games.
>
> > The references section is, frankly, a mess. There is no consistency in the formatting and content of the references and some don't even list a venue, just authors, a title, and a year.
>
> We will update this for the revised version we plan to post in a few days.
>
> > p. 2: "At the time of writing, there has been no successful demonstration of this." It wasn't clear to me at this point what "this" refers to. Even now I'm not totally sure. It would be good to have a more clear expression of the open problem here.
>
> What we meant by “this” is “a learned public belief state belief model”---in particular, a belief model which takes 1) the public observations (the emission variables) and 2) some representation of the players’ policies (the control variables) as input and models the public belief state. The reason this is tricky is that it would seemingly require passing the players’ policy networks into the belief model at each time step. As a result, prior to this submission, there were no scalable methods for producing consistent public belief state models. We believe this is a very important result for the multi-agent RL community, as public belief states were a crucial component of superhuman poker AI.
>
> > All that said, I do have a significant concern about this paper. The empirical results rely heavily on an existing algorithm called RLSearch. As far as I can tell, the paper that describes RLSearch has not been peer reviewed and has only been available on arXiv for about a month at time of reviewing.
>
> Hopefully, we can assuage this concern. The RLSearch paper has been peer reviewed and was accepted to NeurIPS 2021. See link below.
> https://neurips.cc/Conferences/2021/Schedule?showEvent=26648

---

> > ### Comment · Reviewer_kvNa · 2021-11-18
> > **Thanks for the response**
> >
> > All of my significant concerns have been addressed. Thanks! I will edit my review accordingly.
> >
> > (In the future, a good references section could prevent misunderstandings about what has been published...)

---

### Official Review · Reviewer_F3L2 · 2021-11-02

**Correctness:** 4
**Technical Novelty And Significance:** 3
**Empirical Novelty And Significance:** 3
**Recommendation:** 8
**Confidence:** 4

**Main Review:**

This is an interesting paper that is fairly novel, as far as I know.  However, there were some problems in the exposition that left me feeling that I might have misunderstood some aspects of it.  I do think it will be important to clarify some of these points in any published version of the paper.  Many of my confusions were about the types or APIs of various components.  I will start with larger questions, and then list some minor ones.

In section 3.1, I immediately was confused by the type of b^{t+1}_{Y^{t+1}}.   Because Y is capitalized, it seems to be a random variable?  But that doesn't make a lot of sense to me.  Here are some possible interpretations of b^{t+1}_{Y^{t+1}}:
- Given a particular observation Y_{t+1}, then it's just the next belief state (but why is Y capitalized?)
- Not given an observation, it's a random variable over next beliefs b (where the distribution over b is introduced by distribution over observations)
- Not given an observation, it's the next belief state we would get if we made no observation.
- Not given an observation, it's some sort of expectation over the next belief (this is tricky, I think.)

The discussion of the sampling procedure (in that same section) says it produces samples of X^{t+1}---but it seems to be important that we are getting samples of the Y_{t+i}, as well, so we can use them to fine-tune our estimator.

I'm not sure the analogy with approximate dynamic programming is helpful;  or, at least, it would help to clarify it.  In particular, is it critical to actually fine-tune \theta?   An alternative would be to stick with the particle view, and use the sampled X's to represent the posterior belief, or use some combination of the belief produced by the parameteric model \theta and the particles more explicitly, rather than actually changing the \theta.   (To me, it would feel much more like ADP if you were training some function b_\theta to represent the belief-state directly, on each iteration, which would be yet another possible approach I suppose, except that representing belief states is notoriously difficult.)

I really wanted Figure 2 (which is called table 2) clarify things for me, but it really didn't help me at all.   I think something more like a data-flow diagram that illustrates how \theta gets changed over time (and makes very clear that \theta doesn't parameterize B (nothing parameterizes or really explicitly represents B itself), but instead parametrizes the state-estimator "box".) would be more helpful.

The standard error values seem quite small (given the amount of potential variance in the different training processes).  Please state very clearly exactly what sources of variance are being captured here:  initializations / batches /data for training the initial \theta, randomness due to evolution of the game trajectory at runtime (due to card shuffling, randomness in agents' play, etc.), variance of sampling to generate the fine-tuning data, variance of the fine-tuning training (potentially due to batching I guess).   What were the various "N" values?

The performance improvements seem to be real, and interesting, but the gains aren't enormous.  This is fine, but I'm not sure I'd claim that performance is *greatly* improved.

More minor:
- I appreciate the generality of the approach and the desire to be agnostic to the form and training details of f_\theta, but I had to go digging, a bit, into other papers to try to understand the actual approach you used.  A bit more detail about that would have been helpful in understanding your approach.
- In early parts of the paper you talked about going from step t to step t+1;  later it was from step t-1 to step t.
- "which is naively from independence assumptions"
- "covariate distribution should not impact model performance" (missing "shift")

**Summary Of The Paper:**

This paper proposes a strategy for improving a trained parametric model for belief-state update, by generating and training on new data, online.  This strategy is particularly helpful for dealing with problems of covariate shift (when, e.g, the system has moved outside the training regime) and dependence on unobservable factors (e.g. the agent's policies in a game setting).   This fine-tuning strategy is shown both to improve the fidelity of belief-state updates and to have an ultimate impact in the quality of play in Hanabi, a game with a substantial amount of private information.

**Summary Of The Review:**

This paper has interesting ideas that seem to be useful.  Clarity could be improved but interesting enough to publish.  I wanted to give it a 7.

---

> ### Author Response · Authors · 2021-11-18
> **Thanks for the review!**
>
> > I immediately was confused by the type of b^{t+1}{Y^{t+1}}
>
> We agree, this was written in a way that was a bit confusing. We mean the first of your suggestions. We will update this in the revised version of the paper that we will post in a few days.
>
> > but it seems to be important that we are getting samples of the Y_{t+i}, as well, so we can use them to fine-tune our estimator.
>
> True! We will change the language to clarify this point in an updated version we plan to upload in a few days.
>
> > Please state very clearly exactly what sources of variance are being captured here: initializations / batches /data for training the initial \theta, randomness due to evolution of the game trajectory at runtime (due to card shuffling, randomness in agents' play, etc.), variance of sampling to generate the fine-tuning data, variance of the fine-tuning training (potentially due to batching I guess). What were the various "N" values?
>
> Thanks for pointing out this omission. We meant to include this in the original submission. For each experiment we used the same initial \theta for each of the assessed algorithms. The randomness is due to the card shuffling, as well as the sampling used during search and fine-tuning. For the belief quality experiments, we used N = 300 games. For the 5-card decision-time planning experiments, we used N = 2000 games. For the 7-card decision-time planning experiments, we used N = 1000 games.
>
> > The performance improvements seem to be real, and interesting, but the gains aren't enormous. This is fine, but I'm not sure I'd claim that performance is greatly improved.
>
> To clarify, the language “greatly improved” was used in the submission to describe the performance improvements yielded by exact search (not those yielded by the contribution). We stand by the claim that these gains are substantial (the blueprint wins the game 63% of the time whereas multi-agent RLSearch wins the game 76% of the time). That said, we agree “greatly” is not the best word choice and will change the language accordingly.

---

> > ### Comment · Reviewer_F3L2 · 2021-11-26
> > **Thanks for the answers**
> >
> > The clarifications you suggest are helpful.   It might be good to understand whether this works well for a variety of different initial \theta.  But, I remain happy with this paper.

---

### Official Review · Reviewer_Pqjr · 2021-11-03

**Correctness:** 4
**Technical Novelty And Significance:** 2
**Empirical Novelty And Significance:** 3
**Recommendation:** 8
**Confidence:** 3

**Main Review:**


The problem of learning and optimizing over belief states is an important problem in reinforcement learning, and while the ideas in this paper are not particularly novel, the core idea is implemented and tested well. I thought that the lack of details made certain section rocky, but in general found the paper a pleasure to read.

Strengths:

- Simplicity is perhaps the main strength of this paper: the method is very closely inspired by a wide body of existing work on belief filtering, and the method looks like fine-tuning a learned model on augmented data. Empirical details withstanding, the generic idea of re-allocating the "capacity" of a neural network at test-time in this way seems well-justified and simple to implement.

- The paper is easy to read, and exposes the main concepts well.

Weaknesses:

- The paper is a little light on exact algorithmic details (and I did not see more in the appendix). In particular, I think it would be useful to exactly detail out the algorithm being proposed in a more fleshed-out manner somewhere in the main text or appendix. It would also be a good idea within Section 2 to more explicitly what the goal of the belief state modelling problem is, and what the belief state model will be used downstream for (as this dictates what approximations are useful / worthwhile). I found the latter point difficult to understand in the current treatment.
- Along the same lines, I think it would be useful to more explicitly outline in the appendix exactly how BFT is used for down-stream decision-time planning (e.g. with RL Search). To generate an action at a time-step, is BFT (for 10k timesteps) run only once, or is it necessary to run multiple times within the policy search loop?
- I found no details for exactly how the particular choice of 10,000 gradient steps for BFT were chosen. How sensitive is BFT to the number of gradient steps that are taken to fine-tune the learned model? How much does this depend on the quality of the original model: e.g. with a worse original model, does running BFT degrade the quality of the belief since BFT bootstraps off of the original model?
- It seems that this problem assumes access to the "true" underlying state space and the transition dynamics / emission distributions in this state space, in which case the main purpose of the neural network is simply amortization of an exact search problem. It would be useful to make these assumptions more explicit earlier (perhaps even in the introduction). As a question to the authors, how would these techniques transfer to the setting where these assumptions do not hold (in particular, where the underlying state space $\mathcal{X}$ is not known).
- I think the point that the method works because it "refocuses" the capacity of the parametric belief state model is interesting, but I don't think it is supported very strongly by the provided data. It would be interesting to see whether these effects diminish as the neural network capacity increases (or equivalently, if the effects amplify as it decreases), since presumably, the better the original neural network is at the belief modelling problem, the less impact a fine-tuning step should have. Along these lines, I think it would be interesting to have some discussion as to whether fine-tuning procedures can simply be supplanted by training bigger networks.

**Summary Of The Paper:**

This paper seeks to understand whether finetuning a learned belief state model can enable improvements in performance in partially observable problems, in particular in the Hanabi domain. The general intuition is that while the original parametric model is trained on a wide distribution of belief distributions / input histories, during deployment, the model need only be good on the sequence that is actually realized. The authors demonstrate that indeed such techniques lead to improved performance over a non-adaptive belief state method, in particular when there is significant stochasticity in the transitions.


**Summary Of The Review:**

(Copied from above) The problem of learning and optimizing over belief states is an important problem in reinforcement learning, and while the ideas in this paper are not particularly novel, the core idea is implemented and tested well. I thought that the lack of details made certain section rocky, but in general found the paper a pleasure to read.

---

> ### Author Response · Authors · 2021-11-18
> **Thanks for the review!**
>
> > The paper is a little light on exact algorithmic details (and I did not see more in the appendix). In particular, I think it would be useful to exactly detail out the algorithm being proposed in a more fleshed-out manner somewhere in the main text or appendix.
>
> We agree! We will include additional details in an updated version in a few days.
>
> > I think it would be useful to more explicitly outline in the appendix exactly how BFT is used for down-stream decision-time planning (e.g. with RL Search). To generate an action at a time-step, is BFT (for 10k timesteps) run only once, or is it necessary to run multiple times within the policy search loop?
>
> At each decision point, we run BFT one time for 10k gradient steps to improve the belief model’s accuracy for the current time step. After we have improved the belief model, we run RLSearch using the improved belief model for 10k gradient steps. This procedure works because RLSearch only requires the current public belief state. Other examples of search algorithms that only require the current public belief state include SPARTA and Pluribus.
>
> > How sensitive is BFT to the number of gradient steps that are taken to fine-tune the learned model?
>
> We ran new HMM experiments to answer this question. Each result below was aggregated over 300 games.
>
> V0: 2.08 +- 0.01\
> Seq2Seq: 1.68 +- 0.01\
> Seq2Seq + BFT(100 gradient steps): 1.64 +- 0.01\
> Seq2Seq + BFT(1k gradient steps): 1.61 +- 0.01\
> Seq2Seq + BFT(10k gradient steps): 1.61 +- 0.01
>
> BFT yielded benefits even with only a small number of gradient steps. The benefit appears to saturate after 1k gradient steps. This, in part, may be because the belief model is already relatively good in the HMM case. Note that we did not use the same RL/belief networks for these results as for the HMM results in the paper so they are not directly comparable.
>
> > with a worse original model, does running BFT degrade the quality of the belief since BFT bootstraps off of the original model?
>
> We have not observed this effect. Because BFT uses forward bootstrapping (rather than backward), bad models may benefit significantly from BFT. We ran more experiments in the HMM setting to demonstrate this. Each result below is aggregated over 300 games.
>
> V0: 2.08 +- 0.01\
> Seq2Seq(400 epochs): 1.68 +- 0.01\
> Seq2Seq(400 epochs) + BFT: 1.61 +- 0.01\
> Seq2Seq(200 epochs): 1.74 +- 0.01\
> Seq2Seq(200 epochs) + BFT: 1.63 +- 0.01\
> Seq2Seq(100 epochs): 1.82 +- 0.01\
> Seq2Seq(100 epochs) + BFT: 1.70 +- 0.01
>
> BFT produces large improvements even when its original model performs poorly. Note again that we did not use the same RL/belief networks for these results as for the HMM results in the submission so they are not directly comparable.
>
> > It seems that this problem assumes access to the "true" underlying state space and the transition dynamics / emission distributions in this state space, in which case the main purpose of the neural network is simply amortization of an exact search problem. It would be useful to make these assumptions more explicit earlier (perhaps even in the introduction).
>
> Indeed, we do assume access to the dynamics of the system. We mention this in the abstract, background, method, and conclusion (quoted in bullets below).
> - We investigate the challenge of modeling the belief state of a partially observable Markov system, given sample-access to its dynamics model
> - In all cases, we assume sampling access to Y : X → ∆Y and X : X × Z → ∆X.
> - Thus, given sample-access to X , Y, and the distribution of X0
> - The third line is to investigate eliminating the need for sampling access to the dynamics model.
>
> If the reviewer still feels like this has not been adequately expressed then we can work on rewording our statement of this assumption in the paper.
>
> > how would these techniques transfer to the setting where these assumptions do not hold (in particular, where the underlying state space X is not known).
>
> This is an important question! Even in fully observable stochastic settings, this remains an active area of research. For example, a contemporaneous submission to ICLR considers an extension of MuZero to stochastic settings. We believe that the core idea of enforcing belief consistency at decision time will prove valuable, even in a value-equivalent MBRL paradigm. (By value-equivalent, we mean in the sense of “The Value Equivalence Principle for Model-Based Reinforcement Learning”.) Exactly how this would work is not entirely clear yet, but we hope to work toward it in the future.

---

> > ### Author Response · Authors · 2021-11-18
> > **(continued from above)**
> >
> > > I think the point that the method works because it "refocuses" the capacity of the parametric belief state model is interesting, but I don't think it is supported very strongly by the provided data. It would be interesting to see whether these effects diminish as the neural network capacity increases (or equivalently, if the effects amplify as it decreases), since presumably, the better the original neural network is at the belief modelling problem, the less impact a fine-tuning step should have.
> >
> > We agree that it would be interesting to see this effect. We ran more experiments in the HMM setting to investigate. We ran two versions of the experiments. In one version, we fine-tuned the encoder LSTM of the belief model (which is what we did for the results in the submission). In the other, we fine-tuned the entire belief model. The results are aggregated over 300 games.
> >
> > V0: 2.08 +- 0.01\
> > Seq2Seq(LSTM_dimension=128): 1.87 +- 0.01\
> > Seq2Seq(LSTM_dimension=128) + BFT(LSTM): 1.84 +- 0.01\
> > Seq2Seq(LSTM_dimension=128) + BFT(Full): 1.69 +- 0.01\
> > Seq2Seq(LSTM_dimension=256): 1.76 +- 0.01\
> > Seq2Seq(LSTM_dimension=256) + BFT(LSTM): 1.65 +- 0.01\
> > Seq2Seq(LSTM_dimension=256) + BFT(Full): 1.63 +- 0.01\
> > Seq2Seq(LSTM_dimension=512): 1.68 +- 0.01\
> > Seq2Seq(LSTM_dimension=512) + BFT(LSTM): 1.61 +- 0.01\
> > Seq2Seq(LSTM_dimension=512) + BFT(Full): 1.58 +- 0.01
> >
> > It appears to be the case that the effect is amplified as the capacity of the belief model is decreased, but only if fine-tuning is performed on the whole belief network, rather than just the encoder LSTM. Note again that we did not use the same RL/belief networks for these results as for the HMM results in the submission so they are not directly comparable.
> >
> > > I think it would be interesting to have some discussion as to whether fine-tuning procedures can simply be supplanted by training bigger networks.
> >
> > Thank you for this suggestion. We will add discussion on this in the introduction to motivate our approach. While in principle a bigger network and more data could obviate the need for fine-tuning in any RL domain, in practice the amount of extra space and data needed could be prohibitive. Search effectively augments the network size and amount of data by focusing the network and collecting more data for the specific states encountered. This comes at a cost at inference time, but the cost can be minor relative to the gains. As an example, no superhuman AI has been developed for Go that does not use search. Search was also necessary for achieving superhuman performance in games such as backgammon, chess, and poker. In all of these settings, search was in theory unnecessary so long as there was a large enough network and enough training data. Nevertheless, as a matter of practice, search was critical for success in those domains.

---

> > > ### Author Response · Authors · 2021-11-25
> > > **Follow Up**
> > >
> > > Hello reviewer Pqjr, we would be grateful if you can confirm whether our response has addressed your concerns, and let us know if any issues remain. To recap our response, we:
> > > - Added an additional appendix section discussing our implementation
> > > - Added additional experiments investigating the sensitivity of BFT to the number of gradient steps
> > > - Added additional experiments investigating the sensitivity of BFT to the quality of the belief model
> > > - Added additional experiments investigating the sensitivity of BFT to the capacity of the belief model

---

> > > > ### Comment · Reviewer_Pqjr · 2021-11-29
> > > > **Satisfied with response**
> > > >
> > > > Thank you for the response and clarifications! I think that with the additional analyses and experiments, the empirical section of the paper is far more complete and my primary concerns about the paper have been addressed.

---

### Official Review · Reviewer_WyMC · 2021-11-07

**Correctness:** 4
**Technical Novelty And Significance:** 2
**Empirical Novelty And Significance:** Not applicable
**Recommendation:** 3
**Confidence:** 3

**Main Review:**

The abstract, introduction and background is very clearly written. It does a good job of introducing the problem to be solved and outlining the method to be used at a high level.

The method itself is rather mathematically simple. It is based on simple mathematical techniques in the area and effectively amounts to sampling from known distributions.

While the authors state that they present an algorithm, it might be better to view this as a framework, as the method involves an externally supplied model as an input, and the fine tuning method needs to be tailored to this model. The experiments performed use only a single model, which limits the conclusions that can be drawn from the results.

The experimental results are only provided for a single domain, albeit for a HMM, POMDP and FOSG setting in the same model. The choice of domain seems rather poorly selected, as the basic models used as a comparison in the domain already appear to be almost optimal, meaning there is little improvement from the new technique - while the results may be statistically significant, they appear to be a very minor improvements in effect size.

It would be very useful to see some mention of time in the results section. The procedure doesn't appear particularly onerous but in a game playing setting, there may be time limits and any improvement in quality would need to be seen after X seconds, not after X iterations.

These criticisms aside, the experiments appear to be fairly conducted and reported, and look at sensible aspects of the problem, with a good choice of baselines/competitors.



**Summary Of The Paper:**

The authors propose a new method for calculating distributions over states in Markov problems in which the true state is not directly observable. The method relies on using an existing model and then fine tuning the results online at each time step. Experiments presented show this gives a more accurate distribution, and that this more accurate distribution leads to statistically significant improvements in performance.

**Summary Of The Review:**

The authors present an interesting idea well. However, the experimentation is very limited (even for a conference paper) and the results given show a very small effect size. More data is needed to convince that the technique is useful.

---

> ### Author Response · Authors · 2021-11-18
> **Thanks for the review!**
>
> > While the authors state that they present an algorithm, it might be better to view this as a framework, as the method involves an externally supplied model as an input, and the fine tuning method needs to be tailored to this model.
>
> We agree! We will change the wording to better reflect this view.
>
> > The choice of domain seems rather poorly selected, as the basic models used as a comparison in the domain already appear to be almost optimal, meaning there is little improvement from the new technique
>
> We would not characterize the effect as “very minor”. As the joint policy becomes stronger, it becomes increasingly difficult to eke out further expected improvement. For example, for (8 hint, 5 card), the increase in performance from applying BFT on top of seq2seq (24.35 -> 24.58) is far from minor; indeed, this is the difference between winning the game 63% of the time and 74% of the time.
>
> More broadly speaking, one of the goals of this project was to obtain similar results to SPARTA and RLSearch in a more general setting. Given that RLSearch and SPARTA were published (in NeuriPS and AAAI, respectively) largely based on the magnitude of improvement they produced in Hanabi, we do not think the fact that we achieve comparable improvements should be considered a negative.
>
> > It would be very useful to see some mention of time in the results section. The procedure doesn't appear particularly onerous but in a game playing setting, there may be time limits and any improvement in quality would need to be seen after X seconds, not after X iterations.
>
> Needless to say, the answer here is implementation and computational resource specific. For our implementation, using two GPUs (one to generate data and one to perform updates), 10k gradient steps of BFT took 137 seconds per decision point. As a point of reference, 10k gradient steps of RLSearch required 205 seconds on average per decision point with the same resources and implementation. Over the course of a whole game, this is slightly slower than typical human play. Nevertheless, the algorithm scales well, and with greater hardware improvements (or simply more computation) and optimizations the technique will become even faster. Our intention is to develop an algorithm that will be useful in the long term rather than one that is efficient today but will be obsolete within 5 years.
>
> > However, the experimentation is very limited
>
> We have performed additional ablations in response to reviewer Pqjr.

---

> > ### Author Response · Authors · 2021-11-25
> > **Follow Up**
> >
> > Hello reviewer WyMC, we would be grateful if you can confirm whether our response has addressed your concerns, and let us know if any issues remain. To recap, we:
> > - Point out that the precedent in MARL Hanabi literature is to consider the improvements that we observed in the submission as substantial
> > - Added additional experiments investigating the sensitivity of BFT to the number of gradient steps
> > - Added additional experiments investigating the sensitivity of BFT to the quality of the belief model
> > - Added additional experiments investigating the sensitivity of BFT to the capacity of the belief model

---

### Decision · Program_Chairs · 2022-01-20

**Decision:**

Accept (Poster)

**Comment:**

The paper proposes to fine tune the belief states of a MDP, for later using the learned model for decision-time planning, e.g. via search.
The contribution is well-presented, motivated and focused to a specific scenario, which is generally considered challenging in the literature. This scenario is exemplified by the cooperative card game Hanabi, which takes the role of the benchmarking setting for the empirical evaluation of the fine-tuning procedure.

The major concern raised in the review and discussion phases are about the limited evaluation, which is centered around only Hanabi, as well as the magnitudes of the improvements over previous baselines. However, three knowledgeable reviewers agreed that since the setting has been historically challenging, the reported improvements are in fact significant and potentially inspiring future works in this direction.

The paper is accepted provided that the authors include and polish in the camera-ready the additional experiments over the parameter sensitivities, the ablation tests and the discussions highlighted by the reviewers in the comments.